# Phosphatidylinositol Monophosphates Regulate the Membrane Localization of HSPA1A, a Stress-Inducible 70-kDa Heat Shock Protein

**DOI:** 10.3390/biom12060856

**Published:** 2022-06-20

**Authors:** Larissa Smulders, Rachel Altman, Carolina Briseno, Alireza Saatchi, Leslie Wallace, Maha AlSebaye, Robert V. Stahelin, Nikolas Nikolaidis

**Affiliations:** 1Department of Biological Science, Center for Applied Biotechnology Studies, and Center for Computational and Applied Mathematics, College of Natural Sciences and Mathematics, California State University Fullerton, Fullerton, CA 92834, USA; larissa.smulders@age.mpg.de (L.S.); raltman18@csu.fullerton.edu (R.A.); cbriseno89@csu.fullerton.edu (C.B.); ali.saatchi17@csu.fullerton.edu (A.S.); lannewallace@csu.fullerton.edu (L.W.); meyhab@csu.fullerton.edu (M.A.); 2Department of Medicinal Chemistry and Molecular Pharmacology and the Purdue University Cancer Center, Purdue University, West Lafayette, IN 47907, USA; rstaheli@purdue.edu

**Keywords:** cell surface, heat-shock proteins, lipids, pharmacology, plasma membrane, phosphoinositide

## Abstract

HSPA1A is a molecular chaperone that regulates the survival of stressed and cancer cells. In addition to its cytosolic pro-survival functions, HSPA1A also localizes and embeds in the plasma membrane (PM) of stressed and tumor cells. Membrane-associated HSPA1A exerts immunomodulatory functions and renders tumors resistant to standard therapies. Therefore, understanding and manipulating HSPA1A’s surface presentation is a promising therapeutic. However, HSPA1A’s pathway to the cell surface remains enigmatic because this protein lacks known membrane localization signals. Considering that HSPA1A binds to lipids, like phosphatidylserine (PS) and monophosphorylated phosphoinositides (PIPs), we hypothesized that this interaction regulates HSPA1A’s PM localization and anchorage. To test this hypothesis, we subjected human cell lines to heat shock, depleted specific lipid targets, and quantified HSPA1A’s PM localization using confocal microscopy and cell surface biotinylation. These experiments revealed that co-transfection of HSPA1A with lipid-biosensors masking PI(4)P and PI(3)P significantly reduced HSPA1A’s heat-induced surface presentation. Next, we manipulated the cellular lipid content using ionomycin, phenyl arsine oxide (PAO), GSK-A1, and wortmannin. These experiments revealed that HSPA1A’s PM localization was unaffected by ionomycin but was significantly reduced by PAO, GSK-A1, and wortmannin, corroborating the findings obtained by the co-transfection experiments. We verified these results by selectively depleting PI(4)P and PI(4,5)P_2_ using a rapamycin-induced phosphatase system. Our findings strongly support the notion that HSPA1A’s surface presentation is a multifaceted lipid-driven phenomenon controlled by the binding of the chaperone to specific endosomal and PM lipids.

## 1. Introduction

All living cells respond to environmental insults by inducing the expression of heat-shock proteins (HSPs) [1]. Many HSPs are molecular chaperones and are major orchestrators of the cellular stress response [2]. Among the different HSP proteins, the 70-kDa HSPs (Hsp70s) play central roles in protein homeostasis [1]. One of these Hsp70s is the stress-induced HSPA1A, which is overexpressed after stress, e.g., heat shock, and in several pathophysiological conditions, like cancer [3].

HSPA1A exerts its chaperoning and anti-apoptotic functions in the cytosol of stressed and cancer cells [3,4,5]. However, it is now well established that HSPA1A translocates to the plasma membrane (PM) and the endo/lysosomal pathway of cancer and stressed cells [4,6,7,8], and these cells actively export HSPA1A to the extracellular medium (EM) [6,9,10,11,12,13,14,15,16,17]. PM-bound (mHSPA1A) and EM-localized HSPA1A (eHSPA1A) exert immunomodulatory functions and render tumors resistant to standard therapies [3,4,6,7,9,18]. Therefore, manipulating HSPA1A’s PM translocation and EM export are promising therapeutics to reduce tumor resistance and improve health outcomes [19,20].

HSPA1A lacks known PM or EM translocation signals [10,12,21,22,23], and, therefore, its presence in these cellular compartments is enigmatic. To resolve this enigma, several groups, including ours, have independently shown that HSPA1A interacts with specific membrane lipids [6,8,10,15,21,22,23,24,25,26,27,28] and proposed these interactions to explain HSPA1A’s unusual translocation. These lipids include globotriaosylceramide (Gb3), a lipid found primarily in lipid-rafts [8], Bis(monoacylglycero)phosphate (BMP), a lysosomal membrane lipid, and phosphatidylserine (PS), a lipid found primarily in the inner leaflet of the PM during normal cell growth [15,22,23,26,27,28,29]. In all three cases, the interaction between HSPA1A and Gb3, BMP, or PS was critical for HSPA1A’s cell surface presentation [8,10,15,28,30].

In addition to these lipids, experiments using recombinant HSPA1A and artificial liposomal vehicles showed that the protein could bind with relatively high affinity and selectivity to other phospholipids, including cardiolipin, a lipid found exclusively in the eukaryotic mitochondrial membranes, and different phosphoinositides (PIPs) [6,21,22,23,25,26,31]. Although these in vitro findings indirectly support lipid-driven recruitment of HSPA1A to different subcellular compartments, including the PM, these interactions have yet to be shown in cells.

In the case of PIPs, specifically, HSPA1A binds with higher affinity to phosphatidylinositol monophosphates, i.e., PI(3)P, PI(4)P, PI(5)P, than to phosphatidylinositol di- or triphosphates [25]. This finding supports the prediction that if HSPA1A interacts with PIPs inside human cells, inhibiting the interaction with monophosphates will have a greater effect on HSPA1A’s PM localization than inhibition of di- or triphosphates. This prediction is partially supported by experiments in which PI(4,5)P_2_ depletion resulted in minimal loss of PM-localized HSPA1A [21].

Considering the unusual presence of HSPA1A in the endo/lysosomal pathway, the enrichment of this pathway in PIPs, the notion that PI(4)P recruits proteins to vehicles destined for membrane localization, and HSPA1A’s PIP-binding characteristics, we hypothesized that phosphatidylinositol monophosphates regulate the PM translocation and surface presentation of HSPA1A in human cells. To test this hypothesis, we determined how PIPs depletion affects HSPA1A’s PM presence after mild heat shock. We used three approaches to alter cellular PIPs’ content. First, we masked these lipids using co-transfection with known PIP-biosensors (Figure 1). These included the PI(4)P binding protein P4M-SidMx2, the PI(3)P-biosensor EEA1, the PS-biosensor Lact-C2, and the PI(4,5)P_2_-biosensor PLCδ-PH. Second, we used several specific pharmacological inhibitors that decreased PIPs (Figure 1). Specifically, we depleted PI(4,5)P_2_ using ionomycin, PI(4)P using phenyl arsine oxide and GSK-A1, and PI(3)P using wortmannin [32,33,34]. Third, we diminished PI(4)P and PI(4,5)P_2_ using PM localized phosphatases (Figure 1). These enzymes included Sac1, which dephosphorylates PI(4)P into PI, and the inositol polyphosphate 5-phosphatase E (INPP5E), which converts PI(4,5)P_2_ to PI(4)P [34]. In all cases, we quantified HSPA1A in the cytosol or the PM using imaging and biochemistry.

## 2. Materials and Methods

### 2.1. Plasmids Used in This Study

To study HSPA1A’s localization, we used differentially tagged versions of the protein. Specifically, the cDNA clone containing a mouse *hspA1A* gene sequence, accession number BC054782, was used to generate the recombinant clones used in this study. HSPA1A was tagged with green fluorescence protein (GFP), red fluorescence protein (RFP), or myc. For expression in mammalian cells, the gene was subcloned into the pEGFP-C2, mRFP-C1, or pcDNA™3.1/myc-His(-) vectors using directional cloning after PCR amplification and restriction digest (see [22,37] for complete protocol and primers used).

We used several lipid biosensors to selectively remove lipid targets using transfection (Figure 1). Lact-C2-GFP and Lact-C2-mCherry (PS-biosensors) were a gift from Sergio Grinstein (Addgene plasmid # 22852 and # 17274, respectively) [38]. GFP-C1-PLCδ-PH [PI(4,5,)P_2_ biosensor] was a gift from Tobias Meyer (Addgene plasmid # 21179) [39]. GFP-EEA1 wt [PI(3)P-biosensor] was a gift from Silvia Corvera (Addgene plasmid # 42307) [40]. GFP-P4M-SidMx2 [PI(4)P-biosensor] was a gift from Tamas Balla (Addgene plasmid # 51472) [33]. LYN11-FRB-CFP, PJ-Sac1-FKBP-mRFP, PJ-WT-FKBP-mRFP, PJ-INPP5E-FKBP-mRFP, and PJ-Dead-FKBP-mRFP were a gift from Robin Irvine (Addgene plasmid #38003, 38000, 37999, 38001, and 38002, respectively) [32].

### 2.2. Cell Culture, Transfection, and Treatments

To test the location of HSPA1A, we used two different human cell lines. Specifically, Human embryonic kidney cells (HEK293; ATCC^®^ CRL-1573^™^) and Henrietta Lacks’ ‘Immortal’ cells (HeLa; ATCC^®^ CCL-2^™^) were purchased from ATCC (December 2016). These cell lines were maintained in a humidified 5% CO_2_ atmosphere at 37 °C in a complete medium consisting of Dulbecco’s Modified Eagle Medium (DMEM; HEK) or Minimum Essential Media (MEM; HeLa) supplemented with 10% fetal bovine serum, 2 mM L-glutamine, and penicillin-streptomycin (and 0.1 mM non-essential amino acids and sodium pyruvate for HeLa).

We used transient transfection following the protocol below to express the different proteins used in the present study. A day before transfection, cells were split into 24-well plates at 2.0 × 10^4^ cells per well, containing poly-D-lysine treated coverslips, or 75 cm^2^ cell culture flasks at 6.0 × 10^6^ cells per flask. After 18 h, cells were transiently transfected with the appropriate construct using the PolyJet In Vitro PNA Transfection Reagent (SignaGen, Ballenger Creek, MD, USA) as per the manufacturer’s instructions. Transfection was allowed to continue for 18 h, and then the transfection media was removed and replaced with fresh complete media. Cells remained at 37 °C or were placed in a humidified CO_2_ incubator equilibrated at 42 °C for 60 min, followed by recovery at 37 °C for 8 h. For the transfections on coverslips, we used 0.5 μg total DNA (for transfecting two plasmids, 0.25 μg of each plasmid was used, whereas for three plasmids, 0.16 μg per plasmid was used). For the 75 cm^2^ flasks, we used 10 μg of total DNA (for transfecting two plasmids, 5 μg of each plasmid was used).

Following any transfection and subsequent treatment, all cells were fixed in 4% paraformaldehyde solution (PFA) in complete growth medium for 12 min at room temperature (RT). After aspirating the PFA, all cells were washed three times with 1X phosphate-buffered saline (PBS). The cells were incubated with 1 μg/mL wheat germ agglutinin lectin Alexa Fluor^®^ 555 (Thermo Fisher Scientific, Waltham, MA, USA) conjugate for 10 min at RT to stain the PM. After aspirating the PM stain, the cells were washed three times with 1X PBS. All coverslips were mounted to slides with approximately 3 μL DAPI Fluoromount-G^®^ (SouthernBiotech, Birmingham, AL, USA). The slides were allowed to dry for approximately 24 h at RT in the dark and stored at 4 °C until they were visualized using confocal microscopy.

To test the effect of drugs that inhibit the synthesis of specific PIPs (Figure 1) and therefore change the amount and location of these lipids in the cell, we transfected cells with plasmid DNA coding for the expression of HSPA1A-GFP, GFP-C1-PLCδ-PH, EEA1-GFP wt, GFP-P4M-SidMx2, and Lact-C2-GFP. Specifically, we used ionomycin, phenyl arsine oxide (PAO), GSK-A1, and wortmannin.

Ionomycin is a calcium ionophore that, by increasing cellular calcium, activates phospholipase-C (PLC). PLC is a scramblase that causes loss of plasma membrane asymmetry by hydrolyzing the head group of PI(4,5)P_2_ (Figure 1). Long ionomycin treatments will also cause PS externalization to the outer cell surface and reduce PM localized PI(4)P [32,33,34]. In our experiments, transfected cells grown on functionalized coverslips were heat-shocked at 42 °C for one hour or kept at control conditions. The cells were allowed to recover for 8 h and were treated with DMSO or 10μM ionomycin diluted in a buffer containing 10 mM HEPES (pH 7.4), 140 mM NaCl, and 2.5 mM CaCl_2_ for 2 min at room temperature. The cells were then washed three times with PBS, fixed, stained, and mounted on slides as described above.

PAO, a general PI4-kinase inhibitor, causes a significant reduction of PI(4)P at the PM (Figure 1; [32,33,34]). PAO treatment will also result in the reduction of the amount of PI(4,5)P_2_ at the PM but does not affect the amount of PS (Lact-C2) at the PM [33,34,41]. For these experiments, immediately after heat shock and at 7.5 h of recovery, the cell medium was aspirated and replaced with a buffer containing 10 mM HEPES (pH 7.4), 140 mM NaCl, and 2.5 mM CaCl_2_ and DMSO or 10 μM of PAO. The cells were then incubated for 30 min at 37 °C in a humidified CO_2_ incubator. The coverslips were washed three times with PBS, fixed, stained, and mounted on slides as described above.

GSK-A1 is a type III phosphatidylinositol 4-kinase PI4KA (PI4KIIIα) inhibitor that selectively downregulates cellular PI(4)P, but not PI(4,5)P_2_ (Figure 1; [42]). After heat shock and during the 8 h recovery period, the original media were aspirated, and cells were treated with either DMSO or 100 nM GSK-A1 diluted in serum-free DMEM for 30 min at 37 °C in a humidified CO_2_ incubator. The cells were then washed three times with PBS, fixed, stained, and mounted on slides as described above.

Wortmannin is a potent inhibitor of phosphoinositide 3-kinases (PI3Ks) that depletes PI(3)P and derivatives (Figure 1; [32,33,34]). After heat shock and during the 8 h recovery period, the original media were aspirated, and cells were treated with either DMSO or 100 nM wortmannin diluted in serum-free DMEM for 30 min at 37 °C in a humidified CO_2_ incubator. The cells were then washed three times with PBS, fixed, stained, and mounted on slides as described above.

To further delineate whether PI(4)P or PI(4,5)P_2_ are necessary for HSPA1A’s PM translocation, we used a phosphatase expression system to target specifically PI(4)P and PI(4,5)P_2_ at the PM. This system uses the rapamycin inducible dimerization of FKBP domains to recruit enzymes to a PM-anchored FRB domain (Figure 8). These enzymes included the *S. cerevisiae* sac1 phosphatase (Sac1), which dephosphorylates only PI(4)P into PI, the inositol polyphosphate 5-phosphatase E (INPP5E), which converts PI(4,5)P_2_ to PI(4)P, a combination of Sac1 and INPP5E (WT), and an inactive version (Dead) of both enzymes [34]. For these experiments, the control cells were transfected with 0.25 µg of DNA coding for the expression of LYN11-FRB-CFP and one of the following: HSPA1A-GFP, EEA1-GFP wt, GFP-P4-SidM, or GFP-C1-PLCdelta-PH. The experimental cells were transfected with approximately 0.167 µg of DNA coding for the expression of LYN11-FRB-CFP, a GFP-tagged protein, and one of the following: PJ-Sac1-FKBP-mRFP, PJ-INPP5E-FKBP-mRFP, PJ-WT-FKBP-mRFP, or PJ-Dead-FKBP-mRFP. The cells were maintained in a humidified 5% CO_2_ atmosphere and either kept at 37 °C or heat shocked for 1 h at 42 °C and allowed to recover for 8 h at 37 °C. The existing media were aspirated, and the treatment cells were incubated with 1 µM rapamycin diluted in serum-free DMEM for 7 min at 37 °C in a CO_2_ incubator. Non-rapamycin or rapamycin-containing media was aspirated, and cells were washed two times with 1X PBS before being fixed, stained, and mounted as described above.

To determine cell viability after heat shock and other treatments, we used the trypan blue exclusion assay and measured cells with the Cellometer^®^ Auto X4 Cell Counter (Nexcelom Bioscience, Manchester, UK).

### 2.3. Confocal Microscopy and Image Analysis

Cells were visualized using an Olympus FLUOVIEW FV3000 inverted scanning confocal microscope equipped with a 60× 1.5 oil objective. For each multichannel image, raw image files were collected as three channels: channel 1 [green (excitation 488 nm; emission 510 nm)]; channel 2 [blue (excitation 405 nm; emission 461 nm)]; and channel 3 [red (excitation 561 nm; emission 583 nm)].

Image analyses were performed using several cells (the number of cells is provided in the figure legends) from three independent experiments. Images were analyzed manually in ImageJ [43] by the corrected total cell fluorescence (CTCF) method [44] and semi-automated using an in-house MATLAB script (please see https://github.com/LarissaSmulders/ImageAnalysisMatlab.git for code and documentation). The manual procedure followed was the one used in [21,37,45] (see also Appendix A for a graphical representation of the procedure). CTCF was calculated for the plasma membrane and cytosol by the following formula: CTCF = Integrated Density—(Area of Region of Interest x Fluorescence of background reading). A ratio between the CTCF measurements of the PM and the rest of the cell (PM + Cytosol) was then calculated [21,34,37,45]. Because the cell slices used are not completely flat and the method uses the outer pixels of each cell, a value of 0% PM localization cannot be achieved [21,34].

### 2.4. Cell Surface Biotinylation and Total PM Protein Isolation

To further investigate the PM localization of HSPA1A, we performed cell surface biotinylation [21]. For these experiments, immediately following the heat shock, HEK293 cells were trypsinized, washed three times in PBS (pH 8), counted, and incubated in freshly prepared PBS (pH 8) containing 10 mM CaCl_2_ and 1 mg/mL Sulpho-NHS-LC-biotin (Thermo Scientific Pierce) for 30 min at room temperature with constant agitation. The reactions were quenched with 100 mM glycine in PBS (pH 8) for 10 min. Cells were then lysed in 500 μL radioimmunoprecipitation assay buffer (RIPA) buffer, and 500 μg of cell lysate was dissolved in 500 μL of RIPA buffer supplemented with 25 μL of streptavidin agarose beads and rotated at 4 °C for 3 h. Beads were washed with 10 volumes of RIPA buffer, heated at 75 °C in sample buffer (LDS) for 10 min, and analyzed by sodium dodecyl sulfate-polyacrylamide gel electrophoresis (SDS–PAGE) followed by Western blot analysis (see [22] for complete protocol). In these experiments, the loads used for the total protein contained 15 μg of total cell lysate.

The antibodies used were: the OmicsLink™ Anti-GFP Tag Antibody Mouse Monoclonal IgG1 [(CGAB-GFP-0050, Genecopoeia, Rockville, MD, USA); 1:1000- detects a protein band of approximately 30 kDa (empty GFP), two bands of approximately 98 kDa (GFP-HSPA1A and GFP-EEA1), several bands around 52 kDa of the remaining biosensor constructs (Lact-C2-GFP, GFP-P4M-SidMx2, GFP-PLCδ-PH)]; the Anti-c-Myc Antibody (9E10, Genecopoeia, Rockville, MD, USA) [monoclonal IgG (Dilution 1:1000)- detects a protein of approximately 74 kDa (HSPA1A-myc)]; the anti-HSP70 Monoclonal Antibody [(mouse IgG) (Dilution 1:1000) clone #C92F3A-5 (Enzo Life Sciences, Farmingdale, NY, USA; detects a protein around 74 kDa; native and overexpressed HSPA1A)]; THE™ beta actin antibody, GenScript [Piscataway, NJ, USA (mouse mAb; A00702) 1:1000- detects a protein band of approximately 42 kDa; cytosolic loading control]; and the Na+/K+ ATPase α (ATP1A1) antibody RabMAb^®^ [(EP1845Y); (2047-1), Epitomics, Burlingame, CA, USA; 1:1000- detects a protein band of approximately 112 kDa plasma membrane loading control]. All blots were incubated with the antibodies overnight (~16 h) at 4 °C with constant rotation.

The blots were stained for total protein with the Pierce™ Reversible Protein Stain (Thermo Scientific™, Waltham, MA, USA). The Western signals were detected using the Omega Lum™ C Imaging System (Aplegen; San Francisco, CA, USA). The detected protein bands were quantified using densitometry (Image Studio Lite 3.1; Licor Inc., Lincoln, Nebraska), and the ratio between PM (biotinylated fraction)/total (loads) protein intensity was calculated. These values were normalized to controls (control set to 100%), and the standard deviation was scaled accordingly. These experiments were repeated three times.

Finally, to investigate HSPA1A’s membrane localization in addition to embedding, we isolated the total PM proteins from cells expressing HSPA1A, HSPA1A-P4M-SidMx2, and HSPA1A-PLCδ-PH. For these experiments, HEK293 cells were grown to confluency in 75 cm^2^ flasks, and approximately 20 million cells were used per treatment and construct combination. The PM proteins were isolated using the Plasma Membrane Protein Extraction Kit (101Bio, Palo Alto, CA, USA), following the protocol provided by the manufacturer. This non-detergent separation generates nuclear, cytosolic, organelle membrane, and plasma membrane protein fractions. This experiment was performed once.

### 2.5. Statistical Tests

Statistical significance was determined by one-way ANOVA (analysis of variance) with post-hoc Tukey HSD (honestly significant difference) test. A *p* value < 0.05 was considered statistically significant. The boxplots were generated using R software (http://shiny.chemgrid.org/boxplotr/) [46].

## 3. Results

### 3.1. HSPA1A’s PM Localization Is Significantly Reduced by the Presence of P4M-SidMx2 and EEA1

We first verified that HSPA1A’s PM levels increase at the PM at 8 h of recovery after heat shock under our current experimental conditions (Figure 2A upper panels and Figure 2B red box plots). We next determined whether the PM-localization of HSPA1A depends on the presence of PI(4)P and PI(3)P. For this purpose, we selectively blocked HSPA1A’s binding to these lipid targets by co-transfecting HSPA1A with P4M-SidMx2 [33] and EEA1 [32,47], two well-established biosensors for PI(4)P and PI(3)P, respectively. Imaging experiments revealed that in the presence of P4M-SidMx2 and EEA1, HSPA1A’s PM localization was largely unaffected in the control cells (37 °C; Figure 2A,B light and darker blue box plots). In contrast, in heat-shocked cells, the PM-localized HSPA1A was significantly reduced by almost 90% (*p* value < 0.0001), reaching the levels of the control cells (Figure 2A,B light and darker blue box plots).

Next, we determined how the presence of PI(4)P and PI(3)P affect HSPA1A’s embedding to the PM using cell surface biotinylation. For these experiments, we co-transfected GFP-HSPA1A (or HSPA1A-myc) with GFP-P4M-SidMx2 and GFP-EEA1. To control for non-specific effects on HSPA1A’s PM localization due to the presence of any fluorescent protein and equalize the transfected DNA’s amount between the different experiments, we co-transfected GFP-tagged HSPA1A with an empty eGFP-C2 (Appendix A). These experiments revealed that HSPA1A’s PM embedding is increased at 8 h of recovery after heat shock as compared to control cells (Figure 3A anti-GFP panel, Figure 3B anti-myc panel, and Figure 3C graph columns one-two and five-six). Additionally, these experiments revealed that the increase of PM-embedded HSPA1A after heat shock was inhibited by the presence of both P4M-SidMx2 or EEA1 (Figure 3A anti-GFP panel, Figure 3B anti-myc panel, and Figure 3C graph columns three-four and seven-eight). We further evaluated and confirmed the importance of PI(4)P in the heat-induced PM increase of HSPA1A using isolated total PM protein fractions [Appendix A; anti-GFP panels; plasma membrane fractions (P)]. As controls, in parallel experiments, we also included Lact-C2 (PS-biosensor) and PLCδ-PH [PI(4,5)P_2_ biosensor]. This experiment verified that HSPA1A’s PM localization is greatly inhibited by PS, PI(4)P, and PI(3)P blocking but is largely unaffected by PI(4,5)P_2_ blocking (Appendix A; anti-myc panels). Collectively, these experiments establish that PIPs-selectivity is a critical factor for the PM-localization and membrane anchorage and embedding of HSPA1A.

### 3.2. Pharmacological PIP Depletion Reduces HSPA1A’s PM Localization

To confirm the findings that HSPA1A’s PM localization depends on PI(4)P and PI(3)P but not PI(4,5)P_2_, four drugs known to inhibit or activate PIPs lipid metabolizing enzymes were used [34]. These pharmacological compounds were ionomycin, phenyl arsine oxide (PAO), GSK-A1, and wortmannin.

Ionomycin treatment results in a significant reduction of PI(4,5)P_2_ (PLCδ-PH) at the PM (Figure 1). For this reason, in addition to HSPA1A, we also measured the amount of PLCδ-PH at the PM after ionomycin treatment. These experiments revealed that ionomycin treatment significantly decreased (approximately 30%; *p* value < 0.0001) the PM localization of the PI(4,5)P_2_ biosensor PLCδ-PH in cells kept at control conditions as well as in heat-shocked cells (Figure 4A two bottom panels and Figure 4B yellow boxplots). In contrast, the PM localization of HSPA1A was not affected by the ionomycin treatment in either control or heat-shocked cells (Figure 4A two top panels and Figure 4B red boxplots). These findings corroborate previous observations that PI(4,5,)P_2_ minimally affects HSPA1A’s PM localization [21].

PAO, a PI4-kinase inhibitor (Figure 1), known to significantly decrease PI(4)P (and subsequently the P4M-SidMx2) at the PM, was used to further evaluate the importance of PI(4)P in HSPA1A’s PM localization (Figure 5). These experiments revealed that PAO treatment reduced the amount of PI(4)P at the PM (approximately 30%; *p* value < 0.0001) in cells kept under control conditions as well as in cells exposed to heat shock, as shown by the decreased PM localization of the P4M-SidMx2 biosensor (Figure 5A two bottom panels and B light blue boxplots). As expected, PAO does not affect the amount of PS (Lact-C2) at the PM at either condition (Figure 5B orange boxplots). Treatment with PAO, however, revealed that HSPA1A’s PM localization was significantly reduced (approximately 80%; *p* value < 0.0001) in cells exposed to heat shock (Figure 5A top panels and Figure 5B red boxplots). These findings further support the notion that PI(4)P is essential for HSPA1A’s PM localization. Given this finding, we sought to determine whether PAO treatment also affects the embedding of HSPA1A to the PM using cell surface biotinylation (Appendix A; anti-myc panels). These experiments confirmed the imaging results revealing that HSPA1A does not increase at the PM after heat shock in PAO treated cells and further supported the notion that PI(4)P is essential for both localization and embedding of the protein to the PM.

Considering that PAO inhibits all PI4 kinases, we also used GSK-A1 to inhibit only the alpha kinases responsible for the initial steps of PI(4)P production [42]. The results from the confocal image analysis revealed that GSK-A1 treatment significantly decreased (approximately 90%; *p* value < 0.0001) the PM localization of the PI(4)P biosensor P4M-SidMx2 in both control and heat-shocked cells (Figure 6A bottom panels and Figure 6B light blue boxplots). Furthermore, GSK-A1 treatment significantly decreased (approximately 100%; *p* value < 0.0001) HSPA1A’s PM localization after heat shock (Figure 6A top panels and Figure 6B red boxplots). Additionally, these experiments revealed that GSK-A1 treatment significantly decreased (approximately 25%; *p* value = 0.001) the PM localization of the PS biosensor Lact-C2 in both control and heat-shocked cells (Figure 6C bottom panels and Figure 6D orange boxplots). In contrast, the PM localization of the PI(4,5)P_2_ biosensor PLCδ-PH decreased by approximately 3% (*p* value = 0.0045) and only in cells kept in control conditions (Figure 6C top panels and Figure 6D yellow boxplots). Finally, as expected, the GSK-A1 treatment at either control or heat shock temperature did not alter the endosomal localization of the PI(3)P biosensor EEA1 (Appendix A bottom panels).

Wortmannin affects the total PI(3)P levels in the cell, which we can detect using the EEA1 biosensor. This compound was used to further evaluate the importance of PI(3)P in the PM localization of HSPA1A. For this experiment, we used the PI(3)P biosensor EEA1, which is expected to mis-localize to the cytosol as compared to its untreated localization to endosomal vesicles (Figure 7A far right column), as a positive control. We also used the PS, PI(4)P, and PI(4,5)P_2_ biosensors, Lact-C2, P4M-SidMx2, and PLCδ-PH, respectively (Figure 7A second, third, and fourth columns, respectively), all of which are expected to be unaffected by the wortmannin treatment (Figure 7), as negative controls. In all cases, the control experiments provided the predicted results (Figure 7B orange, light blue, and yellow boxplots). Furthermore, these experiments revealed that in the presence of wortmannin the heat-induced PM re-localization of HSPA1A is significantly reduced by approximately 50% (*p* value = 0.001) (Figure 7A first column and Figure 7B red boxplots). We also used cell surface biotinylation and determined that wortmannin treatment affects the embedding of HSPA1A to the PM (Appendix A; anti-myc panel). Together, these observations further support the finding that PI(3)P is important for both localization and embedding of HSPA1A to the PM.

### 3.3. Phosphatase PIP Depletion Reduces HSPA1A’s PM Localization

The results presented above solidified the importance of available PI(4)P for the translocation of HSPA1A to the PM after stress and strongly suggested that PI(4,5)P_2_ plays a less important or no role in this process. To target both lipids at the PM and distinguish between them, we used a rapamycin-induced phosphatase expression system to translocate each respective phosphatase to function at the PM (Figure 8). The Sac1 enzyme converts PI(4)P into PI, and the INPP5E enzyme converts PI(4,5)P_2_ into PI(4)P. The WT construct codes for both Sac1 and INPP5E, whereas the Dead construct codes for inactive variants of both enzymes [32].

Confocal image analysis revealed that the expression of the Sac1 and WT enzymes significantly decreased (approximately 90%; *p* value < 0.0001) the PM localization of HSPA1A after heat shock (Figure 9A,B +PJ-Sac1 and PJ-WT boxplots). In contrast, the expression of the INPP5E (Figure 9B +PJ-INPP5E boxplots) enzyme at the PM did not alter HSPA1A’s translocation to the PM. Both negative controls, Lyn11 and Dead, do not alter the amount of HSPA1A present at the PM at any conditions tested (Figure 9A,B).

Imaging analysis revealed that the expression of the Sac1 and WT enzymes at the PM significantly decreased (approximately 60%; *p* value < 0.0001) the PM localization of the PI(4)P biosensor P4M-SidMx2 in both control and heat-shocked cells (Figure 9C,D +PJ-Sac1 and PJ-WT boxplots). As expected, the expression of the INPP5E enzyme (Figure 9D +PJ-INPP5E boxplots) at the PM resulted in a slight increase of the PI(4)P content in cells kept in both control and heat-shock conditions (approximately 5%; *p* value < 0.0001). Finally, both negative controls, Lyn11 and Dead, did not affect the PM PI(4)P content (Figure 9D).

Confocal image analysis revealed that the expression of the INPP5E and WT enzymes significantly decreased (approximately 25%; *p* value < 0.0001) the PM localization of the PI(4,5)P_2_ biosensor PLCδ-PH in cells kept in control and under heat-shock conditions (Figure 9E,F +PJ-INPP5E boxplots). However, the expression of the Sac1 enzyme at the PM did not alter the PI(4,5)P_2_ content of the PM. Both negative controls, Lyn11 and Dead, also did not change the PM’s PI(4,5)P_2_ content (Figure 9F).

Finally, the induction of Sac1, INPP5E, or WT in either control or heat-shocked cells did not alter the localization of the PI(3)P biosensor EEA1 within the cell (Appendix A).

## 4. Discussion

HSPA1A, a cytosolic molecular chaperone, embeds in the PM of cancer and stressed cells, and these cells actively secrete the protein to the EM. HSPA1A, which lacks canonical PM or EM localization signals [10,12,21,22,23], interacts with specific membrane lipids [6,8,10,15,21,22,23,24,25,26,27,28], and several studies implicated these interactions with HSPA1A’s PM localization and EM export.

In the current report, we demonstrate that the ability of HSPA1A to translocate and anchor at the PM of heat-shocked cells depends on the availability of PM and endosomal monophosphorylated phosphoinositides. We support this conclusion by presenting evidence showing that depletion of PI(3)P and PI(4)P by co-transfection with lipid biosensors, lipid-synthesis inhibitors, and PM-targeted phosphatases significantly reduces HSPA1A’s PM localization and embedding. The results generated using two different cell lines suggest that this phenomenon is not cell-type specific.

Although the ability of HSPA1A to bind to lipids is well-established, the interaction with specific phosphoinositide lipids has been shown to occur only in vitro using recombinant proteins and liposomal vesicles [25]. The results of the current study reveal that these interactions occur in cells and increase our understanding of the lipids regulating HSPA1A’s unconventional PM localization. Specifically, our results reveal that depletion of PI(4)P at the PM using the PM-localized Sac1 phosphatase and masking of PI(3)P at the membrane of early endosomes using EEA1 diminish HSPA1A’s PM localization. These findings strongly support the importance of these two lipids and suggest the significance of the inner PM leaflet and endosomal membrane to HSPA1A’s PM localization. Furthermore, our data confirm that PI(4,5)P_2_ plays a minimal role in the cell surface presentation of HSPA1A under our experimental conditions, as has been previously revealed [21]. The results show that PI(3)P and PI(4)P are required for HSPA1A’s cell surface presentation, whereas PI(4,5)P_2_ is not essential. This finding provides physiological merit to earlier observations that in vitro HSPA1A binds with 10 times higher affinity to monophosphorylated phosphoinositides than to diphosphorylated phosphoinositides [25]. Together, these results suggest that electrostatic interactions, for example, hydrogen bonds between HSPA1A and PI(4,5)P_2_ headgroups, are not sufficient to explain the membrane binding of HSPA1A. Instead, the HSPA1A’s membrane-binding is controlled by selective high-affinity binding to PI(4)P and PI(3)P and depends less on the total lipid or membrane charge.

In our experiments, we used as controls proteins that bind specifically to individual lipids (lipid biosensors). The intracellular localization of these proteins in untreated cells and their relocalization patterns after drug treatments were consistent with the literature [32,33,34]. These results support the validity of the experimental approach and conclusions regarding HSPA1A’s PM localization. The combined interpretation of the different treatments reveals some characteristics of the importance of the lipids that regulate HSPA1A’s PM localization and generate new questions and research avenues. For example, ionomycin treatment results in significant loss of PLC from the PM but does not affect HSPA1A’s PM localization. This finding corroborates other experiments from the current work and the literature. However, it is challenging to explain in-depth because the 2 min ionomycin treatment also results in some loss of PM PI(4)P and PS by externalization, and these lipids play critical roles in HSPA1A’s PM localization. Although currently mere speculation, a possible explanation is that HSPA1A does not depend on a single lipid to go to the PM, and PS externalization might promote HSPA1A’s PM localization as it has been suggested for pre-apoptotic tumor cells [15].

The PAO and GSK-A1 treatments in combination support the importance of available PI(4)P in HSPA1A’s PM localization but present a different challenge that requires some rationalization. PAO does not affect PM localized PS, but after GSK-A1 treatment, membrane PS is reduced by approximately 20%. Although speculative, we propose that this observation might be related to the specificity of GSK-A1, which targets the first reaction generating PI(4)P and affects PS by altering their exchange at the ER-PM contacts [48]. Finally, depletion of PI(3)P using the EEA1 protein as a competitor results in less PM localized HSPA1A than inhibition of PI(3)P synthesis with wortmannin. This result might be explained by the fact that HSPA1A’s PM localization does not depend exclusively on PI(3)P, but also on binding to PS and PI(4)P, both of which remain unchanged by the wortmannin treatment. A different interpretation might be the specific localization of EEA1, the membrane of early endosomes, where it binds PI(3)P [49]. We speculate that masking PI(3)P specifically at the early endosomes has a greater effect on HSPA1A’s PM localization than depleting the lipid throughout the cell. Although the results using the EEA1 biosensor closely follow the literature, the finding that EEA1 is pulled down using streptavidin beads is puzzling. Although EEA1 has been identified in membrane fractions of several different cell types and colocalizes with biotinylated proteins [49,50,51,52], we are unaware of how it could pass through the PM. An unsubstantiated speculation could be that EEA1 is bound to a heavily biotinylated protein, which indirectly results in its inclusion in the biotin fraction.

In addition to using the lipid biosensors as positive and negative controls for the different treatments, the biosensor results from the heat-shocked cells (measured during 8 h of recovery) reveal the biosensors’ localization, concentration, and re-localization patterns. Our current results suggest that the PM localization and CTCF values of Lact-C2, PLCδ, and SIDMx2 remain relatively unchanged between cells grown at 37 °C and cells measured at 8 h of recovery after heat shock. Nevertheless, the heat-shocked cells show slightly higher CTCF values in most cases. By extension, these results suggest that the lipid targets of these biosensors minimally change when measured during recovery from mild heat shock. We must note here that the 8 h recovery is a relatively advanced time during recovery from mild heat shock and corresponds to a major increase in the expression of the native HSPA1A [37,53]. The inclusion of earlier time points will allow us to conclude the effect of heat shock on the cell lipidome.

Our results in the context of the literature support the notion that HSPA1A’s PM localization is a complex lipid-driven phenomenon. Although our current observations do not clarify the specific mechanism by which PS, PI(4)P, and PI(3)P trigger and recruit HSPA1A, these results corroborate particular pathways that have been implicated in the PM translocation or EM export of the protein. Based on these concepts, we speculate that specific and distinct lipid-binding explains why so many different mechanisms have been described to explain PM-localized and EM-exported HSPA1A. For example, we suppose that interaction with PI(4)P, a lipid known to recruit proteins destined for PM localization, and possibly PS could explain the recruitment of the chaperone to the PM. Furthermore, membrane-associated HSPA1A integrates into the plasma membrane and has been found in lipid rafts, and the latter structures have also been implicated in the protein’s EM export [12,54]. These findings could be explained by binding to Gb3 and PS [10,21]. On the other hand, soluble HSPA1A can be secreted via secretory lysosomes [12,55]. Interaction of HSPA1A with the lysosomal lipid BMP or PI(4)P could account for its presence in the lysosomal compartments. HSPA1A can also be released into the EM via extracellular vesicles, including exosomes [56,57]. We speculate that interaction with PS, PI(4)P, and PI(3)P found in early and recycling endosomes could explain the recruitment of the chaperone to these pathways.

Our current results further promote the concept that lipid binding is a specific activity of HSPA1A that is increasing under certain conditions, including stress, cancer, and maybe other disease states. These observations augment our knowledge and expand the list of lipids recruiting HSPA1A to its unusual locations. This information will allow the design of experiments to specifically deplete selected lipids from different parts of the endo/lysosomal pathway and determine the stress-induced or cancer-related changes that trigger HSPA1A’s recruitment to the PM and EM. This knowledge would provide the necessary tools to manipulate HSPA1A’s unconventional PM and EM presence toward sensitizing cancer cells to therapies and activating the immune system against them.

## Figures and Tables

**Figure 1 biomolecules-12-00856-f001:**
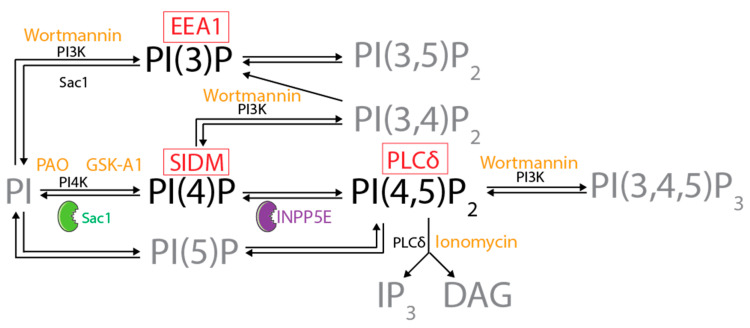
Diagram representing the pathway by which the different phosphorylated forms of phosphatidylinositol are interconverted. The kinase and phosphatase reactions are indicated with black arrows. The two enzymes (Sac1 and INPP5E) used in this study are shown in green and purple. Sac1 dephosphorylates PI(4)P into PI, and inositol polyphosphate 5-phosphatase E (INPP5E) converts PI(4,5)P_2_ to PI(4)P. The inhibitors and drugs used to manipulate specific reactions are indicated in orange on top of the affected enzyme. These included ionomycin that depletes PI(4,5)P_2_, phenyl arsine oxide and GSK-A1 that deplete PI(4)P, and wortmannin that depletes PI(3)P [32,33,34]. The biosensors used to mask specific lipids are shown in red boxes on top of the lipid they each bind. These included the PI(4)P binding protein P4M-SidMx2 (SIDM), the PI(3)P-biosensor EEA1 (EEA1), and the PI(4,5)P_2_-biosensor PLCδ-PH (PLCδ). For convenience, only reactions and lipid species manipulated in this study are shown in black. The information presented stems from [35,36].

**Figure 2 biomolecules-12-00856-f002:**
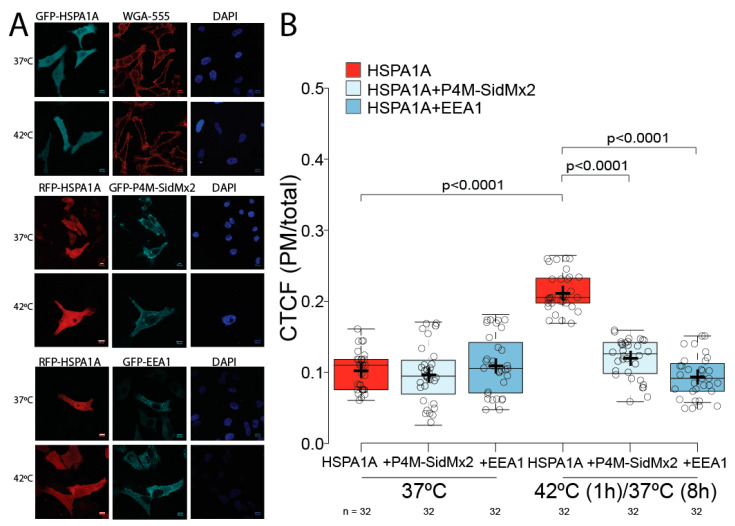
HSPA1A’s PM localization significantly decreases when PI(4)P and PI(3)P are masked by the biosensors P4M-SidMx2 and EEA1. (**A**) Representative images of HeLa cells expressing GFP-HSPA1A stained with WGA-FA555 PM stain and DAPI nucleus stain (top panels). Representative images of the co-expression of RFP-HSPA1A with GFP-P4M-SidMx2 (middle panels) and RFP-HSPA1A with GFP-EEA1 (bottom panels). The localization of HSPA1A was documented for all transfections at either control conditions (37 °C; top row of each panel) or heat-shocked conditions (1 h at 42 °C followed by 8 h at 37 °C; bottom row of each panel). Scale bar = 10 μm. (**B**) Quantification of the corrected total cell fluorescence (CTCF) as a ratio between the total HSPA1A fluorescence of the PM and the rest of the cell at control conditions and after heat shock in the presence or absence of lipid biosensors. The experiment was repeated three times, and the total number of cells (shown as open circles) per condition are shown at the bottom of the graph. The center lines show the medians; box limits indicate the 25th and 75th percentiles as determined by R software; whiskers extend 1.5 times the interquartile range from the 25th and 75th percentiles; crosses represent sample means.

**Figure 3 biomolecules-12-00856-f003:**
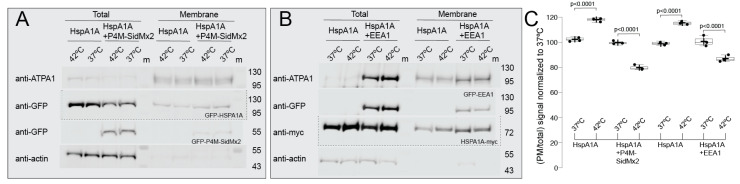
Cell surface biotinylation reveals that HSPA1A’s PM embedding significantly decreases in the presence of biosensors masking PI(4)P and PI(3)P. Representative cropped Western blots showing the total and biotinylated fractions of HEK293 cell lysates transfected with (**A**) GFP-HSPA1A and eGFP (see Appendix A for complete blots), and GFP-HSPA1A and GFP-P4M-SidMx2, (**B**) HSPA1A-myc and eGFP (see Appendix A for complete blots), and HSPA1A-myc and GFP-EEA1. HSPA1A is shown in (**A**) at the anti-GFP panel and in (**B**) at the anti-myc panel. Native actin and ATP1A1 were used a cytosolic and membrane controls, respectively. P4M-SidMx2 is visible in (**A**) at the bottom anti-GFP panel and EEA1 at the anti-GFP panel of (**B**). M: molecular size marker (Fisher BioReagents™ EZ-Run™ Prestained Rec Protein Ladder; approximate sizes shown on the left side of the blots). (**C**) Quantification of the antibody detected signals of the HSPA1A (GFP or myc tagged) in the absence (graph columns one-two and five-six) or presence (graph columns three-four and seven-eight) of P4M-SidMx2 or EEA1 presented as a ratio between the biotinylated (PM) fraction and the total cell lysate. Densitometry values are averages of three independent experiments (n = 3). These values were normalized to controls (control set to 100%), and the standard deviation was scaled accordingly. Center lines show the medians; whiskers extend 1.5 times the interquartile range from the 25th and 75th percentiles; crosses represent sample means.

**Figure 4 biomolecules-12-00856-f004:**
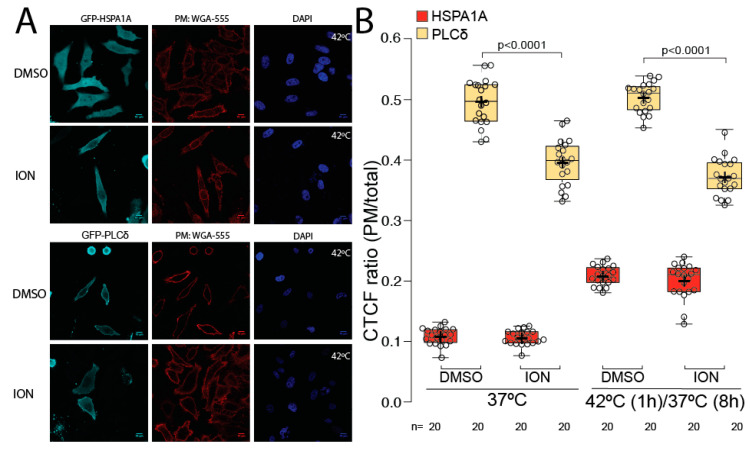
Plasma membrane localization of HSPA1A is not affected by ionomycin. (**A**) Representative images of HeLa cells expressing GFP-HSPA1A (two top panels) and GFP-PLCδ (two bottom panel). Cells were either incubated with DMSO (untreated controls; first row of each panel) or treated with 10 µM ionomycin for 2 min (second row of each panel). Only heat-shocked cells are shown. The PM of all cells was stained with WGA-FA555 and the nucleus with DAPI (second and third picture of each panel, respectively). Scale bar = 10 μM. (**B**) Quantification of the corrected total cell fluorescence (CTCF) as a ratio between the total fluorescence at the PM and the rest of the cell at control conditions and after heat shock in the presence or absence of ionomycin. The experiment was repeated three times, and the total number of cells (shown as open circles) per condition are shown at the bottom of the graph. The center lines show the medians; box limits indicate the 25th and 75th percentiles as determined by R software; whiskers extend 1.5 times the interquartile range from the 25th and 75th percentiles; crosses represent sample means.

**Figure 5 biomolecules-12-00856-f005:**
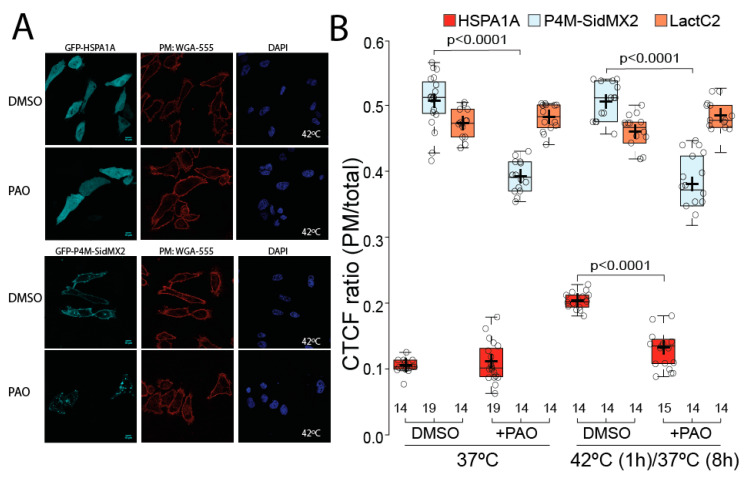
Phenyl arsine oxide (PAO) treatment significantly decreases HSPA1A’s PM localization after heat shock. (**A**) Representative images of HeLa cells expressing GFP-HSPA1A (top panel) and GFP-P4M-SidMx2 (second panel). Cells were either left untreated (DMSO; first row of each panel) or treated with 10nM PAO for 30 min (bottom row of each panel). Only heat-shocked cells are shown. For all cells, the PM was stained with WGA-FA555 and the nucleus with DAPI (second and third picture from the left of each panel). Scale bar = 10 μM. (**B**) Quantification of the corrected total cell fluorescence (CTCF) as a ratio between the total fluorescence at the PM and the rest of the cell at control conditions and after heat shock in the presence or absence of PAO. The experiment was repeated three times, and the total number of cells (shown as open circles) per condition are shown at the bottom of the graph; box limits indicate the 25th and 75th percentiles as determined by R software; whiskers extend 1.5 times the interquartile range from the 25th and 75th percentiles; crosses represent sample means.

**Figure 6 biomolecules-12-00856-f006:**
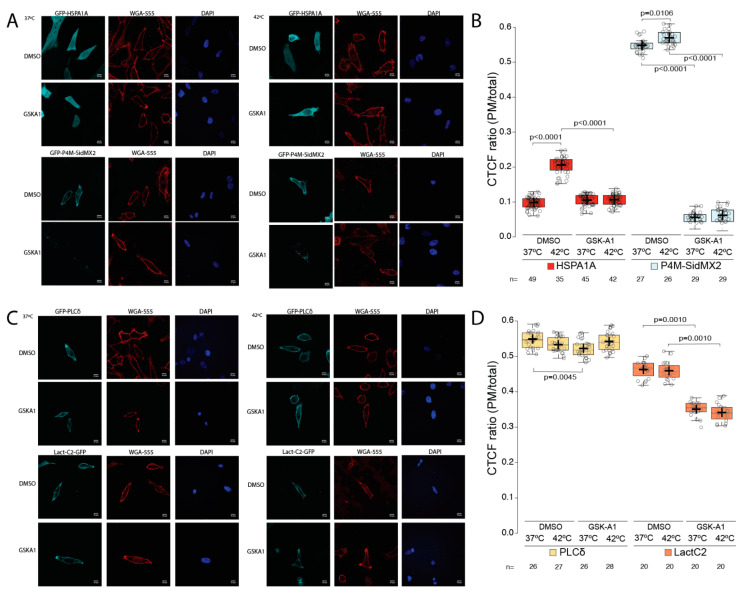
HSPA1A localization at the plasma membrane is significantly reduced by treatment with GSK-A1. Representative images of HeLa cells expressing (**A**) GFP-HSPA1A (top two rows), GFP-P4M-SidMx2 (bottom two rows), (**C**) GFP-PLCδ-PH (top two rows) and Lact-C2-GFP (bottom two rows). In all cases, the PM was stained with WGA-FA555 and the nucleus with DAPI. The PM localization of the proteins was documented at either control conditions (37 °C; left three columns) or heat-shocked conditions (1 h at 42 °C/8 h at 37 °C; right three columns). Negative control cells were treated with DMSO. Scale bar = 10 μM. (**B**,**D**) Quantification of the corrected total cell fluorescence (CTCF) as a ratio between the total GFP fluorescence of HSPA1A, P4M-SidMx2, PLCδ-PH, or Lact-C2 at the PM and the rest of the cell at control conditions and after heat shock in the presence of either DMSO (untreated) or GSK-A1. The experiment was repeated three times, and the total number of cells (shown as open circles) per condition are shown at the bottom of the graph. The center lines show the medians; box limits indicate the 25th and 75th percentiles as determined by R software; whiskers extend 1.5 times the interquartile range from the 25th and 75th percentiles; crosses represent sample means.

**Figure 7 biomolecules-12-00856-f007:**
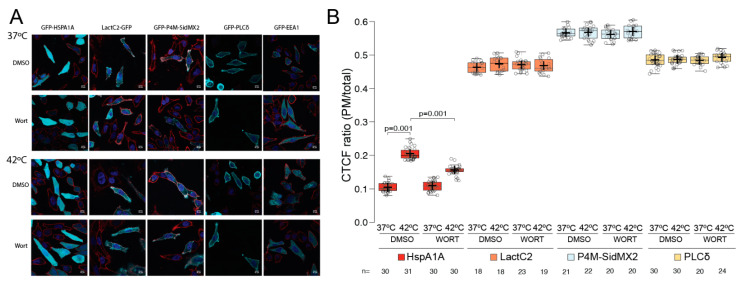
HSPA1A localization at the plasma membrane is significantly reduced by treatment with wortmannin. (**A**) Representative images of HeLa cells expressing GFP-HSPA1A (first column from the left), Lact-C2-GFP (second column), GFP-P4M-SidMx2 (third column), GFP-PLCδ-PH (fourth column), and GFP-EEA1 (fifth column). In all cases, the PM was stained with WGA-FA555 and the nucleus with DAPI. The PM localization of the proteins was documented at either control conditions (37 °C; top panels) or heat-shocked conditions (1 h at 42 °C/8 h at 37 °C; bottom panels). Negative control cells were treated with DMSO. Scale bar = 10 μM. (**B**) Quantification of the corrected total cell fluorescence (CTCF) as a ratio between the total GFP fluorescence of HSPA1A, Lact-C2, P4M-SidMx2, or PLCδ-PH at the PM and the rest of the cell at control conditions and after heat shock in the presence of either DMSO (untreated) or wortmannin. Please note that the EEA1 cells were not quantified because they do not localize at the PM. The experiment was repeated three times, and the total number of cells (shown as open circles) per condition are shown at the bottom of the graph. The center lines show the medians; box limits indicate the 25th and 75th percentiles as determined by R software; whiskers extend 1.5 times the interquartile range from the 25th and 75th percentiles; crosses represent sample means.

**Figure 8 biomolecules-12-00856-f008:**
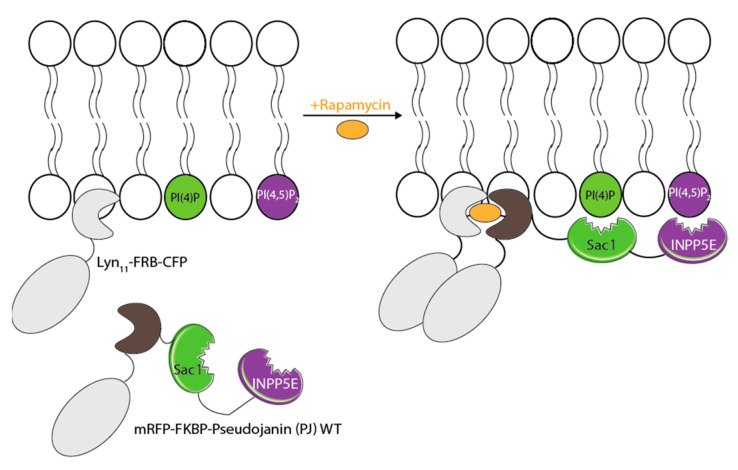
Schematic depicting the principle of the lipid depletion assay using phosphatases. The Pseudojanin (PJ) construct generates a fusion of Sac1 and INPP5E phosphatase domains with FKBP (mRFP-FKBP-PJ-WT). After rapamycin treatment, the mRFP-FKBP-Pseudojanin (PJ) WT construct is recruited to the plasma membrane (PM) due to the dimerization of its FKBP domain to a PM targeted FRB domain (Lyn_11_-FRB-CFP). The information presented stems from [32].

**Figure 9 biomolecules-12-00856-f009:**
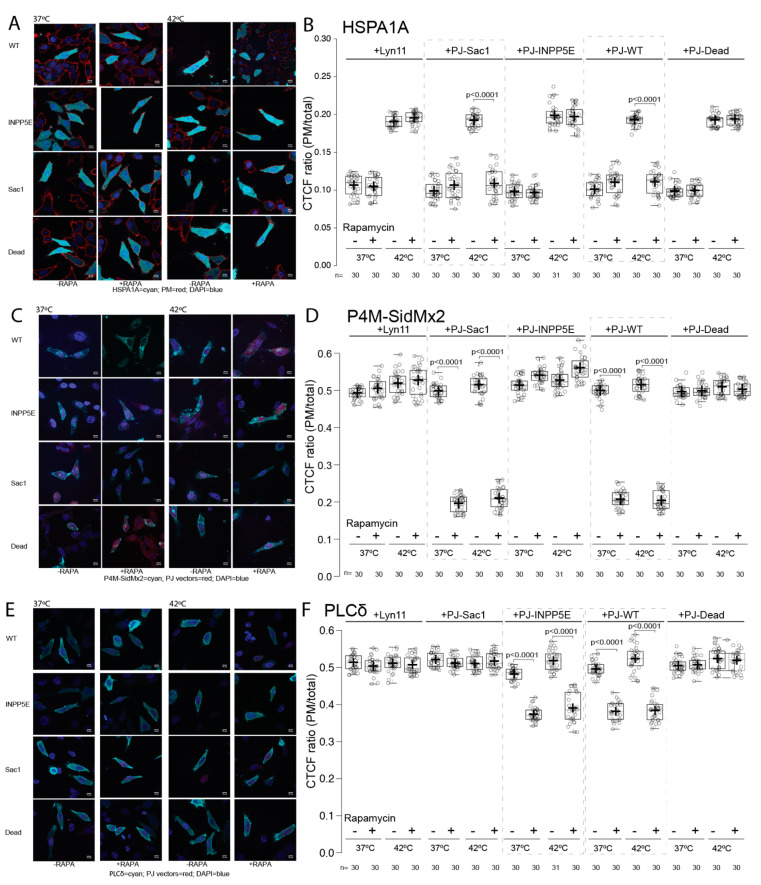
HSPA1A localization at the PM is decreased by the expression of Sac1 and WT enzymes. Representative images of HeLa cells expressing (**A**) GFP-HSPA1A, (**C**) GFP-P4M-SidMx2, and (**E**) GFP-PLCδ-PH with RFP-tagged PJ vectors and DAPI nucleus stain. The PM localization of HSPA1A was documented for all transfections at either control conditions (37 °C; left columns) or heat-shocked conditions (1 h at 42 °C/8 h at 37 °C; right columns). Negative control cells contain either the vehicle alone (Lyn11) or the inactive vector (Dead). Scale bar = 10 μM. (**B**,**D**,**F**) Quantification of the corrected total cell fluorescence (CTCF) as a ratio between the total GFP fluorescence of the protein of interest at the PM and the rest of the cell at control conditions and after heat shock with or without rapamycin treatment. The experiment was repeated three times, and the total number of cells (shown as open circles) per condition are shown at the bottom of the graph. The center lines show the medians; box limits indicate the 25th and 75th percentiles as determined by R software; whiskers extend 1.5 times the interquartile range from the 25th and 75th percentiles; crosses represent sample means.

## Data Availability

All data reported are provided in the text or in Appendix A.

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
