# Peer review of "Phosphatidylinositol Monophosphates Regulate the Membrane Localization of HSPA1A, a Stress-Inducible 70-kDa Heat Shock Protein"

_biomolecules, 2022, doi:10.3390/biom12060856_

Round 1

Reviewer 1 Report

The work appears very interesting, since it adds new knowledge on the mechanisms responsible for the plasma membrane translocation of HSPA1A, which seems to be regulated by PIPs interaction. These, in turn, could be useful to develop novel anti-cancer therapeutic approaches targeting the HSPA1A-PIPs interaction and thus the protein exposure to the surface of both cancer and stressed cells, since plasma membrane-bound HSPA1A was found to exert immunomodulatory functions, and render tumor resistant to standard therapies.

However, in my opinion, the manuscript, especially the description of the obtained results, can be improved. Below are some suggestions and advice.

Minor revisions

Content

Lines 42-44: Please add pertinent citations
Lines 46-47: Please add pertinent citations
Lines 48-49: Please add pertinent citations
Lines 66-69: Please add pertinent citations

Lines 89-90: I suggest describing here which inhibitors have been used, their targets, and the mechanism of action, i.e. how they reduce the amount of the targeted PIP, instead to describe them only in the “Materials and Methods” and “Results” sections (lines 154-158 and 344-346, for the ionomycin; lines 164-165 and 365-369, for PAO; lines 171-172 and 396-397, for GSK-A1; lines 177-178 and 410-411, for Wortmannin). In fact, it would be more useful to provide detailed information about these inhibitors in the Introduction, together with the corresponding references, and give some key information about them in the “Materials and Methods” and “Results” sections, which, in my opinion, should mainly describe the used procedures and performed experiments, and the obtained results, respectively.

Lines 90-91: Again, I suggest describing also here, in a more detailed manner, the used phosphatases, their targets, and the mechanism of action, as well as the corresponding references, and give some key information, useful to understand the performed experiments, in the “Materials and Methods” section.

Lines 94-99, Figure 1 legend: I really appreciate the adding of a Figure which schematically illustrates the different pathways involving the tested PIPs, with the corresponding inhibitors and phosphatases. However, to facilitate the understanding of the figure, I suggest describing the shown pathways and reactions also in the legend and not only in the main text. In this manner, the reader could easily find in the legend the key points to understand what the Figure shows, instead of looking for the detailed description in the main text.

Lines 268-270 and lines 501-504: In my opinion, these preambles are unnecessary and obvious. Please, consider removing them from the text.

Lines 271-276: Again, In my opinion, this preamble is unnecessary, since it states again what has been said in the Introduction, where the authors already clarified the rationale and the aim of the work. Please, consider removing it.

Lines 279-280: For the benefit of clarity, I suggest to indicate the figure/image specifically showing this result.

Generally, for the benefit of clarity, I suggest indicating exactly which image (images of the panels shown in Figures 2A, 4A, 5A, 6A, 6C, 7A, 9A, 9C, 9E, and/or column of the graphs shown in Figures 2B, 4B, 5B, 6B, 6D, 7B, 9B, 9D, 9F) shows each described result, instead of randomly indicating the figure. In this manner, it can be easier for the reader to find it and verify what you state.
The same for the result corresponding to Figures 3A, 3B and 3C.
For instance, for the results shown in figure 2 and described in lines 285-290, I suggest describing carefully the results, highlighting the differences among the various compared conditions. Moreover you should indicate which images (images of the panel shown in Figure 2A and columns of the graph shown in Figure 2B), specifically show each obtained and discussed result.

Lines 328-333: In my opinion, the figure legend should only describe what the figure shows. Therefore, it is unnecessary providing here all this information, which was already given in the dedicated paragraph of the “Materials and Methods” section in the main text.

Typos
Line 342: “(Johnson et al., 2016)”. Please, adjust the reference citation as all the other ones provided in the main text.

Author Response

We thank the reviewer for their constructive comments. Please see our response in the attached file.

Reviewer 2 Report

The Authors aimed to show that HSPA1A plasma membrane level increases after heat shock, and that masking of PI(4)P and PI(3)P inhibits this translocation. They used several methods to manipulate the conversion between different phosphatidylinositol forms and several biosensors to mask (?) specific PIs. Translocation was studied 8h after HS. It is not clear to me why the authors stated that the PM localization of HSPA1A is maximal 8h after HS (line 271). In the cited publication, also only one experimental point was shown.

An increased HSPA1A membrane level is not visible on representative microscopy images, so,  I have to believe that quantification (based on already published protocols) is correct. I am not also convinced by the results shown on western blots (e.g. in Figure 3). Increased PM-embeded HSPA1 after HS may simply reflect an increased total HSPA1 level. The levels of PM-embeded HSPA1A on western blot seem to be higher in the presence of biosensors that mask PI(4)P and PI(3)P (regardless of the temperature). How to explain this result? EEA1 is known to bind both PI(3)P and Rab5 and is a critical factor required for endosome fusion. P4M domain of the SIDM protein binds selectively to PI(4)P (and according to previous papers, showed its localization in the Golgi and the plasma membrane and Rab7-positive late endosomes/lysosomes). Do these biosensors actually mask PI(4)P and PI(3)P? Perhaps their overexpression accelerates rather than inhibits transport to the cell membrane.

Figure 3 legend contains unnecessary details (the antibodies are described in the methods) but there is no information as to why ATPA1/ATP1A1 (discrepancy between the description in the figure and the legend) is shown. I also have to guess how to interpret anti-GFP staining. Lines 307-312: “For these experiments, we co-transfected GFP-HSPA1A (or Myc-HSPA1A) with GFP-P4M-SidMx2 and GFP-EEA1 to verify that the presence of these domains inhibits the PM-localization of HSPA1A. To control for non-specific effects on HSPA1A's PM localization due to the presence of any fluorescent protein and equalize the transfected DNA's amount between the different experiments, we co-transfected GFP-tagged HSPA1A with an empty eGFP-C2.” More description is necessary directly in the Figure (e.g. for detection of SIDM). eGFP from the empty vector was shown only in the supplement (Fig. S1) and its level was higher after HS. How to interpret this?

The abstract should be corrected. “These experiments revealed that depletion of PI(4)P and PI(3)P by co-transfecting HSPA1A with known lipid-biosensors” – suggests that HSPA1A transfection leads to PI depletion. From the description of the experiments with ionomycin, phenyl arsine oxide, GSK-A1, and wortmannin, it should be clear that PI(3)P and PI(4)P are involved in the translocation of HSPA1A to the membrane.

GFP-EEA1 wt (line 112) or EEA1-GFP wt (line 192) – which one was used? The introduction of a GFP tag at the COOH terminus of EEA1 results in the gross enlargement of early endosomes in vivo, coinciding with a loss of specificity for PI(3)P binding. Generally, the authors should have consistent labels. It seems that GFP is used randomly (at N-terminus or C-terminus), also in other fusion proteins (HSPA1A was subcloned into the pEGFP-C2, so EGFP-HSPA1A fusion protein was obtained, not HSPA1A-EGFP). Different labels in different Figures are also used for SIDM.

Accession number BC054782 was used to generate the recombinant clones. Information that it is sequence coding for mouse HSPA1A should be given.

Lines 119-125:  HEK293 and HeLa were used, but growing conditions are given for HeLa and HepG2.

I do not like the CTCF acronym used here for corrected total cell fluorescence because it stands for CCCTC-binding factor.

Please remove lines 268-270 and 501-504.

Figure 2A: why magnification differs between images?

Figures 6 and 9 (different parts) are inappropriately cited in the results section.

Figure 7 – “EEA1 cells were not quantified because they do not localize at the PM”. GFP tagged EEA1 was detected at the plasma membrane as shown in Figures 3 and S1B.

Why do supplementary Figures have no legend?

Author Response

(The authors gave the same response as above.)

Round 2

Reviewer 2 Report

The authors have provided an extensive rebuttal to my previous comments and have mostly addressed my previous concerns. However, the manuscript still needs some editorial work to verify that everything is correct after approval of the changes. For example, there is still an inconsistency in Figure 2A (RFP-HspA1A or HspA1A-RFP?).

Author Response

We thank the reviewer for their satisfaction with our answers. We revised figure 2 and checked the manuscript again for inconsistencies. Any changes are now included in the R2 version of the manuscript.